# Novel Biopolymer-Based Catalyst for the Multicomponent Synthesis of *N*-aryl-4-aryl-Substituted Dihydropyridines Derived from Simple and Complex Anilines

**DOI:** 10.3390/molecules29081884

**Published:** 2024-04-20

**Authors:** Giovanna Bosica, Roderick Abdilla

**Affiliations:** Green Synthetic Organic Chemistry Lab, Department of Chemistry, University of Malta, 2080 Msida, Malta; roderick.abdilla.12@um.edu.mt

**Keywords:** *N*-aryl-4-aryl-substituted dihydropyridines, piperazine–agar, biopolymer-based heterogeneous catalyst, multicomponent reactions, union of MCRs, spiro-linked dihydropyridines, green, recyclable, cheap, metal-free

## Abstract

Although Hantzsch synthesis has been an established multicomponent reaction method for more than a decade, its derivative, whereby an aniline replaces ammonium acetate as the nitrogen source, has not been explored at great length. Recent studies have shown that the products of such a reaction, *N*-aryl-4-aryldihydropyridines (DHPs), have significant anticancer activity. In this study, we successfully managed to synthesize a wide range of DHPs (18 examples, 8 of which were novel) using a metal-free, mild, inexpensive, recoverable, and biopolymer-based heterogeneous catalyst, known as piperazine, which was supported in agar–agar gel. In addition, 8 further examples (3 novel) of such dihydropyridines were synthesized using isatin instead of aldehyde as a reactant, producing spiro-linked structures. Lastly, this catalyst managed to afford an unprecedented product that was derived using an innovative technique—a combination of multicomponent reactions. Essentially, the product of our previously reported aza-Friedel–Crafts multicomponent reaction could itself be used as a reactant instead of aniline in the synthesis of more complex dihydropyridines.

## 1. Introduction

In recent decades, the concept of multicomponent reactions (MCRs) has percolated into synthetic organic chemistry because of green and environmental considerations [1]. Reductions in reaction time, purification length, and solvent use, as well as the ability to synthesize final complex products that cannot be formed otherwise, are some of the inherent advantages of using MCRs. These can be catalysed by various methods, such as by the use of magnetic nanocatalysts or aqueous micelles [2,3,4]. In relation to this, dihydropyridines (DHPs) have long been synthesized via the classical Hantzsch MCR synthesis using a broad range of catalysts, deep eutectic solvents, ionic liquids, microwave/ultrasound radiation, etc. [5,6]. Without doubt, such advances could be driven further because of the significant biological activity exhibited by these scaffolds, as is attested by their incorporation into drugs including nifedipine, amlodipine, and oxodipine (all of which are antihypertensive and antianginal) [7]. Furthermore, a recent publication by our research group showed that *N*-unsubstituted DHPs exhibit significant anti-proliferative activity against various human cancer cells [8].

Notwithstanding the fact that the classic Hantzsch synthesis has been extensively researched, work on the related synthesis, whereby the nitrogen source is an aromatic/aliphatic amine (instead of ammonia/ammonium acetate), has been given relatively less attention. That said, there are studies which suggest that *N*-aryl-4-aryldihydropyridines (Figure 1) demonstrate significant anticancer activity. Specifically, sulfonamide-bearing *N*-aryl-substituted dihydropyridines are highly active because they are able to inhibit the activity of carbonic anhydrase isoenzyme (catalyses cellular conversion of carbon dioxide into bicarbonate) [9,10].

One approach to synthesizing the above structures or similar ones necessitates the combination of arylamines (**1**), aldehydes (**2**), dimedone/1,3-cyclohexandione (**3**), and malononitrile (**4**), as portrayed in Figure 1. In truth, one can also arrive at the same final product (**5**) by combining a previously formed sample of the condensation product of the amine and dimedone/1,3-cyclohexandione (the enaminone, structure **I**, Figure 2) and the Knoevenagel product of the reaction between the aldehyde and malononitrile (the arylidene malononitrile, structure **II**, Figure 2), as reported in a study performed in DMSO [11].

The current state of the art for the synthesis of the product scaffold shown in Figure 1 includes the use of magnesium oxide nanoparticles [12], zinc oxide nanoparticles (in presence of ultrasound radiation) [13], magnetite [14], triazine diamine-functionalised silica-coated manganese ferrite [15], DBU (in the presence of microwave radiation) [16], molecular sieves (in DMSO solvent, two-component version) [11], chitosan (two-component version) [17], [18-C-6Na][3-NO_2_-C_6_H_4_-SO_3_] crown ether [18], hexamethylenetetramine-based ionic liquid anchored onto MIL-101 [19], and tetrabutylammonium fluoride (two-component version) [20]. Yet, most of these catalysts suffer from one or more disadvantages, especially in terms of the catalyst’s nature and synthesis. In fact, a significant number of them are metal-based, while others require complex hazardous synthetic pathways. Additionally, there have been few attempts at using other functionalised anilines. In light of this, we are reporting the use of the biodegradable, cheap, and easily synthesizable metal-free catalyst piperazine after its incorporation into agar. Such a novel catalyst was discovered to be fitting for the mentioned dihydropyridines, as well as for other MCR studies that are ongoing.

In addition to the above, the aldehyde can be replaced by isatin (**6**) to form spiro-linked *N*-aryl-4-aryl DHP compounds (**7a**), as portrayed in Figure 2 and as reported by Lee, Y.R. et al., who performed the reaction in tetrahydrofuran solvent, using ethylene diacetate as an ionic salt [21]. Positively, in our study, the developed catalyst could be used to form various isatin-derived dihydropyridines in ethanol solvent in appreciable yields.

## 2. Results and Discussion

### 2.1. Catalyst Screening and Condition Optimisation

For the model reaction required to identify the optimum conditions, aniline (**1a**)**,** benzaldehyde (**2a**), dimedone (**3a**), and malononitrile (**4a**) were chosen as the reactants. In an elaborate set of catalyst screening and condition optimisation trials (Table 1), it was immediately evident that it was ideal to first stir at 85 °C in the neat conditions **1a** and **3a** to form the enaminone, before then adding the catalyst and the solvent, **4a** and **2a** (in that order). This was because, when the reactants had been added contemporarily along with the catalyst, the main product that formed was the 4*H*-benzopyran, produced from the combination of dimedone, malononitrile, and the aldehyde (Figure 3) [22]. In fact, the formation of the latter was one of the recurring issues in the optimisation runs, especially because of the fact that during recrystallization both the expected product (**5a**) and the aforementioned side product (**8a**) precipitated out.

Considering that the previous research completed by our group had identified piperazine supported on Amberlyst^®^ 15 as an ideal catalyst for the combination of salicylaldehyde, malononitrile, and nucleophilic species, the latter catalyst was tried in the first couple of screening runs [23]. Increasing the reaction time of the second part of the reaction was evidently beneficial (Table 1, **entry 1** vs. **entry 3**), and it was also ideal to leave dimedone and aniline to react for long enough before adding the remainder of the reactants (Table 1, **entry 1** vs. **entry 2**). Increasing the catalyst amount (**entry 13**) or adding molecular sieves (Table 1, **entry 10**) both helped to improve the yield further. However, there was still significant room for yield improvement.

Replacing piperazine with either ethylenediamine or morpholine as the supported base on Amberlyst^®^ 15 did not help to resolve the issue (Table 1, **entries 8**, **9**). Furthermore, upon replacing the acidic support (Amberlyst^®^ 15) with Dowex^®^ 50Wx8-50-100, the yield values remained fairly similar (Table 1, **entries 10**–**16**). However, when the aldehyde and malononitrile equivalents were increased slightly, the apparent yield increased to 82%. ^1^H NMR analysis later confirmed that the increase was due to the enhanced side-product (**8a**) formation, which precipitated out along with the product.

Aminopropylated cellulose (Cell-NH_2_) performed very poorly, possibly due to the lower basicity of the primary amine group, as did Amberlyst^®^ A26 and Amberlyst^®^ A21 (Table 1, **entries 17**–**20**). The latter two are significantly basic (the former polymer has hydroxide ions ionically bonded to quaternary ammonium centres, whilst the latter has tertiary amine groups), and as such were expected to give appreciable yields. Yet, being spherical mesoporous beads, their surface area in contact with the reactants was relatively small. In addition, polymeric resins are often poisoned during the reaction, as could in fact be inferred from their change in colour. Metallic oxides (supported and free) all fared inadequately (Table 1, **entries 21**–**30**), mainly because of a significant amount of side-product (**8a**) formation. The supported heteropoly acids did not perform better either, which was possibly because the Knoevenagel condensation step occurs preferentially in basic conditions (Table 1, **entries 31**–**32**).

Interestingly, when nanomagnetite was synthesized in situ within montmorillonite K30 and then used as a catalyst (Table 1, **entries 33**–**40**), there was a significant improvement in yield, despite the fact that malononitrile was used in slight excess (1.2 equivalents) in all of the latter trials. In truth, the presence of the benzopyran side product (**8a**) could be noted in all collected products (**5a**) of the said entries (as per ^1^H NMR), and hence alternative catalysts had to be probed. Positively, the novel catalyst piperazine supported within agar (Pip–Agar) provided a very pure final product at the highest yield yet (80%) obtained when used at a molar concentration of 25% (Table 1, **entry 44**). Lower loading still provided a very pure final product (Table 1, **entry 45**), albeit at a lower yield, whereas the use of excess malononitrile and benzaldehyde was detrimental to final product purity because of benzopyran side-product (**8a**) formation. In an additional trial performed using only piperazine (25 mol%) as catalyst, significant side-product formation was observed. This side product precipitated out along with the product. Contrastingly, when untreated agar (0.3 g) was tried as a catalyst (entry not shown), with time it swelled up. As such, stirring could not ensue, even on adding ethanol solvent (+2 mL), and product formation was insignificant. Alternative amines or ionic liquids dispersed within the agar matrix gave average results when used as catalysts (Table 1, **entries 46**–**48**).

### 2.2. Expanding the DHP Substrate Scope

Having determined the ideal conditions (**entry 44**, Table 1), these were adopted for substrate screening. When the aldehyde was varied, yields ranged from 49% to 80% (**entries 1**–**7**, Table 2), with the lower values obtained from the electron-rich 4-methoxybenzaldehyde and the heteroaromatic thiophene-2-carbaldehyde. The former aldehyde furnished an electron-rich Knoevenagel product (after reacting with malononitrile), which was less likely to be then attacked by the enaminone and form the final product. Meanwhile, although in principle thiophene-2-carbaldehyde and benzaldehyde are relatively similar in terms of aromatic character, the former gave an inferior yield, mainly due to significant side-product formation (several minor spots could be observed on TLC plates during reaction monitoring). 

The use of various substituted anilines furnished a set of novel compounds, with yields ranging from average to very good (**entries 8**–**12**, Table 2, 45–84%). Impressively, even the azo dye 4-(phenylazo)phenylamine yielded the final product (**entry 15**, Table 2, 54%), as did the other azo dye derived from 3-methylaniline (3-methyl-4-(3-methylphenylazo)phenylamine) (**entry 16**, Table 2, 23%). Aliphatic amines such as *c*-hexylamine and benzylamine (**entries 13**, **14**, Table 2) performed poorly, with the former furnishing minute traces and the latter giving a yield of just 10%. Other aliphatic amines were explored (not shown in table), and no products could be collected in a pure form.

Replacing malononitrile with methyl cyanoacetate furnished the expected product (**entry 8**, Table 2) at an inferior yield (25%), which might be attributed to the lower α-hydrogen acidity and hence lower reactivity of methyl cyanoacetate with aldehyde. In addition, the resulting Knoevenagel product for attack by the enaminone (originating from the condensation of dimedone and aniline) is less electron-poor. Replacing dimedone with indanedione (**3b**) resulted in a complex reaction mixture with several side-product formations, and no expected product was collected (**entry 17**). On the other hand, cyclohexanedione (**3c**) (**entries 18**–**19**, Table 2) furnished the expected pure products, despite the fact that it was observed that in both instances the reaction mixture turned significantly dark over the course of the reaction, which was possibly due to product decomposition.

Upon substituting the aldehyde with isatin (**6**) to form various spiro-linked dihydropyridines (**7**), the yields were comparable to those reported in the literature (Table 3) [21]. However, one needs to point out that, to the best of our knowledge, this reaction has never been reported as having been performed in a heterogeneous fashion or under green conditions (the reported study used THF as the solvent) [21]. Furthermore, although at first sight the yields obtained when using 3- or 4-methylaniline (**entries 2** and **7**, Table 3) appeared odd, such values can be easily explained as follows. During the course of the reaction, isatin dimerized to form a deep-violet solid (the dimer was confirmed by ^1^H NMR and gave a purple spot-on TLC) that tended to precipitate out with the product during recrystallization from ethanol. The products derived from other amines were poorly soluble in acetone and ethanol and hence, upon washing the final collected products with acetone, the dimer could be removed while retaining the product. On the other hand, the products derived from the methyl-substituted anilines dissolved appreciably in acetone and hence upon washing a fraction of the products **7b** and **7g** ended up being lost. Regrettably the azo-linked aniline **1h** did not furnish the expected product **7h** in pure form due to side-product formation.

### 2.3. Catalyst Description and Recycling Runs

#### 2.3.1. Optical Microscopy

The catalyst was prepared by first dissolving agar (2.0 g) (with heating), a natural polysaccharide of agarobiose with minor peptide components, in water (20 mL). Then, piperazine (3.0 mmol, 0.26 g) was duly added to the mixture and stirred well, and we then left it to set. Subsequently, once the formed gel had hardened, it was manually broken down (Figure 3) into small fragments (using a metal grater) before being heated and dried in an air oven set at 80 °C for 12 h (Figure 4). At this stage, the fragments had hardened and could be broken down into smaller shards via mechanical action (pestle and mortar).

The final catalyst was observed to be transparent shards when examined using optical microscopy (Figure 5i,ii). This confirmed that there were no aggregate formations and that piperazine was trapped in a dissolved state within the matrix.

SEM images (Figure 5iii,iv) helped to understand the size distribution (1–250 μm) and the irregular rough surface that the catalytic particles have. Smaller fragments are adsorbed to larger ones, and no particularly porous structure is noted. This results from the catalyst’s method of preparation (the drying of an aqueous gel). 

#### 2.3.2. IR Spectrum

When comparing the IR spectrum of agar (Figure 6 and Figure 7) with that of pip–agar, one swiftly notes the large OH stretching band at 3418 cm^−1^ for the latter due to the presence of water within the catalyst fragments. This band most certainly overlaps with the N-H stretching vibrations of the piperazine moieties. At 2181 cm^−1^, the pip–agar spectrum has a small peak that could be a result of a piperazine N-H-coupled stretching mode. In both spectra, the bands at 1651/1653 cm^−1^ are possibly an indication of carbonyl stretching caused by carboxylate moieties of the peptide components of agar, whereas the bands at 1456/1458 cm^−1^ and 1373/1377 cm^−1^ are likely caused by aliphatic C-H bending modes. The strong band at 1074/72 cm^−1^ is likely to derive from the C-O stretch of alcohol groups of the agarobiose rings. Other bands below 1000 cm^−1^ are a result of aliphatic C-H bending modes.

In order to confirm the loading of piperazine within the catalyst fragments, a specific mass of the latter was weighed and then transferred, with stirring, to boiling, deionised water in order to redissolve it. The resulting solution was titrated with hydrochloric acid using phenolphthalein as indicator (details in Section 3) and alkalinity was always found to be between 1.00 to 1.10 mmol/g.

#### 2.3.3. Recycling Runs and Hot Filtration Test

Although the catalyst had several green aspects in terms of toxicity and preparation, the most important factor with respect to environmental benignity was its ability to be recycled easily and reused. In fact, considering its physical nature, catalyst recoverability was very simple, and filtration could be performed without difficulty and quickly as no filter pore blockage took place (as happens when filtering fine catalysts).

After leaving the recovered catalyst to dry to a constant weight in an air oven at 80 °C, it could be reused for up to 4 consecutive runs (Figure 8) for the model reaction, yielding **5a**. The drop in yield could be attributed to minimal piperazine leaching or to catalyst poisoning, as the brown-deepening colour change suggested.

For the heterogeneity test, after mixing dimedone (**3a**) and aniline (**1a**) for 7 h to form the enaminone (as per usual procedure), malononitrile (**4a**) and benzaldehyde (**2a**) were added in that order, followed by the catalyst. Initially, the mixture was stirred in neat conditions before it thickened (after about 20 min). At this point, ethanol (2 mL) was added and, considering the physical nature of the catalyst (pip–agar), the reaction mixture could be easily decanted into another clean reaction flask and left to undergo stirring for 8 further hours at the original temperature of 85 °C. After purifying the reaction mixture, the product was collected at a paltry yield of 40%. In addition, TLC analysis during the course of the reaction did not elucidate any changes in the relative intensity of the enaminone intermediate and of the final product. The ^1^H NMR of the crude reaction mixture indeed showed an approximately 56% conversion rate (on comparing a singlet area of the enaminone to that of the product). All of this showed that the reaction stopped proceeding once the catalyst was removed, hence confirming that piperazine was significantly stable and did not leach over the course of the reaction.

### 2.4. Green Metrics of Model Reaction

The atom economy (AE), the E-factor, and the process mass intensity (PMI) for the model reaction (resulting in the formation of **5a**) were calculated as tabulated in Table 4. The high atom economy is an intrinsic positive aspect of the reaction since the only by-product is water (two equivalents), while the *E*-factor is low due to minimal solvent use and a completely recoverable catalyst. Meanwhile, the total process mass intensity calculated below also takes into consideration the amount of solvent used during purification (approximately 20 mL of ethanol, needed for recrystallization) in order to provide a complete picture [24].

### 2.5. DHPs Derived from the Products of the Aza-Friedel–Crafts Multicomponent Reaction

As a further confirmation of the vast potential of multicomponent reactions in synthetic chemistry and of the functional group selectivity and catalytic ability of the catalyst conceived of, it was decided to try to synthesize *N*-aryl-4-aryldihydropyridines derived from another multicomponent reaction. To this effect, considering the free amino group in the final products, the indole derivatives depicted in Figure 4 were synthesized by following our published method for the aza-Friedel–Crafts (AFC) reaction [25].

Thereafter, the same procedure described in Section 2.1 for the synthesis of 1-aryl-4-aryldihydropyridine (**5**) was followed to obtain the complex product (**12**, Figure 5), fully characterised by IR, MS, ^1^H NMR, ^13^C NMR, DEPT135, ^1^H-^1^H COSY NMR, ^1^H-^13^C HSQC, and HMBC NMR (see Appendix A). To the best of our knowledge, this is a completely novel approach to synthesizing highly functionalized *N*-aryl-substituted DHPs and could prove to be a significant breakthrough in the history of the reaction. More so, such products or similar ones (using other derivative aldehydes, indoles, or amines) could potentially demonstrate high biological activity.

### 2.6. DHP Characterisation by Single and Two-Dimensional NMR Methods

The *N*-aryl DHP **5b** was completely characterised by various NMR experiments (apart from IR and MS) to confirm its structure as well as peak assignments beyond reasonable doubt. Product **5b** was specifically chosen (rather than model product **5a**) due to the resolvable ^1^H NMR peaks that allowed for the performance of two-dimensional NMR experiments.


*^1^H NMR spectrum of **5b***


In the ^1^H NMR spectrum (Appendix A), the recognisable pair of doublets at 7.24 and 7.12 ppm can be directly attributed to H2 and 6 and H1 and 5, respectively (the latter pair of hydrogens being *ortho* to the methyl group and hence rendered slightly more electron-rich than the former pair via a positive inductive effect). In the *N*-aryl moiety, H20–22 appear in the multiplet at 7.62–7.55 ppm, whereas H18 and 19 give a doublet-like peak at 7.33–7.28 ppm, evidencing that in this case the predominant factor affecting the electron–density distribution is its proximity to the electron-rich dihydropyridine moiety and the positive mesomeric effect exerted by the nitrogen.

A characteristic singlet at 4.72 ppm confirms the presence of H7, whilst the slightly broad peak at 3.99 ppm is because of hydrogens 25 (-NH_2_). Meanwhile, the methyl hydrogens (H29) appear at 2.31 ppm. Hydrogens 13 and 16 appear as a pair of doublets at 2.20 and 2.14 and at 2.04 and 1.81 ppm, respectively. The latter observation can be explained in terms of the geminal coupling of each distinct equatorial or axial hydrogen. The assignment of the pairs is confirmed below via ^1^H-^13^C HMBC. H27 and 28 also give two separate singlets at 0.96 and 0.85 ppm because the ring moiety to which they are bonded is restricted from fast axial–equatorial equilibration, rendering them non-chemically and magnetically equivalent.


*^13^C, DEPT 135, ^1^H-^13^C HSQC, ^1^H-^13^C HMBC NMR spectra of product **5b***


In order to confirm completely the structure of **5b,** apart from the usual ^13^C and DEPT135 NMR spectra, two other experiments were conducted: a ^1^H-^13^C HSQC, showing direct C-H correlation, and a ^1^H-^13^C HMBC, showing long-range C-H correlations (all spectra available in Appendix A).

The easily attributable peak at 195.67 ppm is due to the carbonyl carbon atom C15, the peak of which is absent in DEPT135. C11 and C10 (both absent in DEPT135 spectrum) are likely to appear at 150.12 and 149.17 ppm, respectively, with the former being deshielded by two nitrogen atoms and the latter by one nitrogen atom only. In addition, they are *sp^2^*-hybridized (hence significantly deshielded compared to other C atoms). Further upfield, C3 (absent in DEPT135, confirmed via ^1^H-^13^C HMBC, as per the explanation below) is visible at 142.77 ppm. The two remaining quaternary aromatic carbon atoms that appear slightly further upfield are C4 and C17, which appear at 136.36 and 136.08 ppm, respectively (both absent in DEPT135). C22 would be the peak at 130.60 ppm, as can be understood from the cross-peak obtained in the HSQC spectrum.

C20 and 21 and C18 and 19 appear at 129.79 and 129.28 ppm, respectively (as per the presence of cross-peaks in ^1^H-^13^C HSQC). Meanwhile, C1 and 5 and C2 and 6 give peaks at 129.28 and 126.99 ppm, respectively, as per the ^1^H-^13^C HSQC spectrum. It is interesting to note that the attached hydrogens show an inverted chemical-shift relationship, with H2,6 being more deshielded than H1,5. This shows that the electron–density distribution in carbon atoms is less influenced by hyperconjugation effects, and more by electron-densities in its vicinity (the dihydropyridine ring). C23, being involved in a triple bond as well as aromatic carbons, is not deshielded and therefore appears upfield at 120.98 ppm. C8, absent in DEPT135 and in the vicinity of the carbonyl group (experiencing some of the negative inductive and ring-current effects), ought to be present at 113.32 pm, whereas C9 is much further upfield because it is shielded by the ring-current effects of the triple-bond system. C13 and C16 appear at 50.03 and 41.71 ppm, respectively (the former is *sp^3^*-hybridized but close to the carbonyl group). C7, an aliphatic carbon atom, ought to give rise to the peak at 35.64 ppm considering the DEPT spectrum and the fact that it is *sp^3^*-hybridized and more distant from the carbonyl group than C13. Lastly, C27 and C28 appear at 32.50 and 29.45 ppm, respectively, as two peaks of approximately equal heights, whereas C29 (rising slightly higher than previous two peaks) is the most shielded carbon atom at 21.11 ppm. The ^1^H-^13^C HMBC spectrum confirms several aspects of the above description by the presence of cross-peaks between long-range-coupled systems based on carbon protons:(1)C15 is long-range coupling with H13 only, hence confirming the above methylene assignment. Simultaneously, C10 and C11 are both coupled to H16, but not to H13.(2)H20 and 21 are long-range-coupled with C17, but H18 and 19 are not. This may seem strange but, in some instances (especially in *para*-substituted systems with chemically equivalent hydrogens on either side of the aromatic ring), the HMBC quantum filter removes two-bond correlations (along with single-bond correlations), leaving only three/four-bond correlations. In fact, C3 gives a cross-peak with H2 and 6 (three-bond correlation), while it does not with H1 and 5. In addition, there are two peaks, which are denoted as 20–21 and 18–19 due to three-bond correlations (H20-C21 or H21-C20 for the former and H18-C19 or H19-C18 for the latter).(3)H2 and 6 and H1 and 5 both give cross-peaks with C2,6 and C1,5, respectively. This may appear incorrect, but what is actually happening is a correlation between H2 and C6 or between C2 and H6 (three bond correlations), similar to what is observed between H/C 20–22.(4)H1 and 5 give a cross-peak with C4 (three-bond correlation). The identity of C4 is confirmed because it undergoes correlation with H7. Meanwhile, C3 is correlated to the methyl hydrogens H29 and with H2 and 6, as already mentioned.(5)H7 carries out long-range coupling with both C10 and C11, with C10 also carrying out long-range coupling with H16 (confirming the assignments of methylene hydrogens). H7 also carries out long-range coupling with C9, C8, C23, C4, C2, and 6.(6)H29s are able to give cross-peaks with C1 and 5 (three-bond correlation).(7)C15 carries out long-range three-bond coupling with H13 (visible cross-peak in expanded inset image), hence confirming the earlier assignments of the methylene hydrogens.(8)C8 is involved in long-range coupling with H7 (strongly), H16 (strongly), and with H13 (barely visible).(9)C9 is involved in long-range coupling with H7 (two-bond correlation) and H25 (three-bond correlation). The two-bond correlation between C9 and H7 is detected because of a small *J* coupling constant between the two that allows for peaks not to be filtered out by quantum filtering.(10)C23 is involved in long-range coupling (three-bond), with H7 confirming its identity.

## 3. Experimental

### 3.1. General Reaction Procedure

Dimedone (1.25 mmol) and aniline (1.25 mmol) were added in that order to a nitrogen-flushed dry 25 mL two-necked flask and stirred in neat conditions at 85 °C for 7 h. In cases when the magnetic stirring bar stopped rotating, 2–3 drops of ethanol (dry) were added. Subsequently, the catalyst (25 mol% of pip–agar, 0.28 g), malononitrile (1.50 mmol), and the aldehyde (1.25 mmol) were added, and a mixture was left to stir at 85 °C until the reaction mixture solidified (for the spiro-linked DHPs isatin (1.25 mmol) was used instead of the aldehyde). Then, ethanol (2 mL) was added. The reaction was monitored using TLC at 1–2 h intervals, and this was stopped until the complete consumption of the enaminone intermediate or until no further changes were observed. Subsequently, the reaction mixture was dissolved in hot acetone or ethanol, filtered using a G4 funnel, and then concentrated under vacuum conditions via rotary evaporation. The crude solid was recrystallized from ethanol (approximately 20 mL required).

For the complex dihydropyridine **12** derived from aza-Friedel–Crafts products, the latter were synthesized by following our reported procedure [25] before then employing the pure aza-Friedel–Crafts products in the same manner as outlined above (replacing the aniline). Hence, the aza-Friedel product in question (1.25 mmol) was stirred at 85 °C along with dimedone (1.25 mmol) in neat conditions (4–5 drops of ethanol added to aid stirring) for 8 h, before we then added the catalyst, malononitrile, and the aldehyde.

### 3.2. Pip–Agar Catalyst Preparation and Alkalinity Determination

Agar (2.0 g) was dissolved in boiling deionised water (20 mL) and stirred until the resulting mixture started to thicken slightly. Subsequently, piperazine (3 mmol) was added and the mixture was stirred for a further 5 min before being decanted into a shallow 250 mL petri dish and left to cool for 1 h in a refrigerator. Then, the hardened gel was broken down into small fragments using a metallic grater, before being heated in an air oven at 80 °C for 6–8 h until they started to turn brown in colour.

The same procedure was followed when infusing agar with DABCO, ethylenediamine diacetate, or hexamine.

In order to calculate the alkalinity, 0.1–0.2 g of the catalyst fragments was boiled in water (15 mL) until complete dissolution. This was then titrated with 0.02 M HCl, using phenolphthalein as an indicator, until we reached a transparent end point.

### 3.3. Other Catalyst Preparations

*Piperazine/Morpholine (Morph)/Ethylenediamine (EDA) supported on Amberlyst 15**^®^ or Dowex^®^ 50Wx8-50-100*
[23]

We added 0.6 g of Amberlyst 15^®^, previously dried at 100 °C in an air-oven for 12 h, to a nitrogen-flushed dry 50 mL single-necked flask, followed by 1–2 mL of ethanol solvent. Subsequently, 0.4 g of the amines (piperazine, DABCO, morpholine, and ethylene diamine) was added to the flask. The mixture was left to stir for 48 h at RT, after which it was filtered through a G4 funnel. The solid catalyst was heated further for 12 h at 100 °C in an oil bath, and we then noted the mass difference compared to that of Amberlyst 15^®^. Finally, the catalyst was transferred to a pestle and mortar and ground into a fine powder.

The same procedure was followed when Dowex 50Wx8-50-100^®^ was used as the acidic support.


*Cellulose–NH_2_ (Cell-NH_2_)*


Aminopropylated cellulose was prepared by following the method reported in [26].


*Magnesium oxide, MgO-SiO_2_, MgO-MCM41*


Nano-magnesium oxide was obtained by following the method reported in [27]. MgO-SiO_2_ was simply prepared via grinding a defined amount of nano-magnesium oxide in a pestle (for 15 min) with silica or nanoporous silica MCM41.

MCM41 was prepared by following the method reported in [28].


*CaO–Boehmite*


Boehmite was synthesized by following the method reported in [29]. Nano-CaO was also synthesized by following a reported procedure [30]; then, a specific amount of the two was ground together in a pestle for 15 min.


*ZnO–Cellulose*


Nano-zinc oxide was synthesized by modifying a method in the literature [31]. Essentially, nano crystalline cellulose (0.9399 g) was stirred at 700–800 rpm in water (50 mL) at room temperature, followed by the addition of 0.137 g (zinc (II) acetate dihydrate)). Subsequently, NaOH solution (0.33 g NaOH in 72 mL water) was added dropwise and then stirred overnight before being filtered and washed with water several times and then left to dry in a desiccator.


*SnO_2_-MK30*


Nano tin (IV) oxide was synthesized after executing the method reported in [32]. Thereafter, a specific amount of tin (IV) oxide was ground in a pestle (for 15 min) along with an amount of Montmorillonite K30.


*Nano-Fe_3_O_4_-MK30*


The catalyst was prepared by carrying out the method reported in [33].


*WSi/PW-MK30*


Both catalysts were obtained by completing the method described in [34].

### 3.4. Product Characterization Procedure

IR spectra were recorded on a Shimadzu IRAffinity-1 FTIR spectrometer (Kyoto, Japan) calibrated against 1602 cm^−1^ polystyrene absorbance spectra. After carrying out a background scan using KBr only, the samples were analysed as KBr pellets. The pellets were prepared by grinding about 5–10 mg of each separate sample with 100 mg of oven-dried potassium bromide with a pestle and mortar, before subjecting the samples to pressure in a screwable die. The final spectra were given as % transmittance against wavenumber (cm^−1^) and could be analysed and processed by the software IRsolution^® ®^ ver. 1.10 before being exported as .txt files and then opened in MS Office Excel^®^.

The NMR spectra were recorded on a Bruker Avance III HD^®^ NMR spectrometer, (Billerica, USA), equipped with an Ascend 500 11.75 Tesla superconducting magnet and operating at 500.13 MHz for ^1^H and 125.76 MHz for ^13^C, and a multinuclear 5 mm PABBO probe. Samples were dissolved in deuterated chloroform, DMSO, or acetone (with TMS). For ^1^H NMR, the product (3–5 mg) was dissolved in 0.8 mL of deuterated solvent, whilst for ^13^C NMR, DEPT135, ^1^H-^13^C HSQC, ^1^H-^1^H COSY, and ^1^H-^13^C HMBC spectra, the mass of product (dissolved in the same volume) was increased to 25–30 mg. The NMR spectra were analysed and processed using Topspin Software, ver. 3.2^®^, and MestreNova^®^, v12.0.2.

Mass spectra were performed using a Waters Acquity^®^ TQD system, (Milford, MA, USA), equipped with a tandem quadrupole mass spectrometer, and analysed directly with a probe. The spectra were obtained in relative abundance compared to *m*/*z* and were generated by the software MassLynx^®^, ver. 4.2The melting points of products were determined using a Griffin^®^ melting point determination apparatus (London, UK) fitted with a mercury thermometer. Three separate readings were taken, and the mean average was then calculated to achieve better accuracy.

Optical microscopy images were obtained using a Nikon Optiphot-100 light optical microscope (Tokyo, Japan) equipped with a Leica DFC290 digital camera (Wetzlar, Germany).

Scanning electron microscopy (SEM) images were obtained by analysing gold-spluttered samples adsorbed to double-sided carbon conducting tape fixed to a metallic stub. The instrument used was a cool-stage CoXem instrument (Daejeon, South Korea) with a tungsten electron source and an acceleration voltage that could vary from 1 to 30 kV. The detector could detect both secondary electronsand four sectors’ back-scattered electrons.

### 3.5. Product Characterization Data

**[5a]** *2-amino-7,8-dimethyl-5-oxo-1,4-diphenyl-1,4,5,6,7,8-hexahydroquinoline-3-carbonitrile* [14]. White solid. M.P. 238–240 °C. IR (KBr) (ν, cm^−1^): 3333, 3221, 3055, 3022, 2959, 2901, 2868, 2180, 1653, 1595, 1572, 1491, 1418, 1373, 1261, 1146, 1042, 831, 741, 698. ^1^H NMR (500 MHz, Chloroform-*d*) δ 7.66–7.56 (m, 3H), 7.41–7.29 (m, 6H), 7.25–7.18 (m, 1H), 4.78 (s, 1H), 4.05 (s, 2H), 2.23 (d, *J* = 16.3 Hz, 1H), 2.17 (d, *J* = 16.3 Hz, 1H), 2.07 (d, *J* = 17.5 Hz, 1H), 1.83 (d, *J* = 17.7 Hz, 1H), 0.98 (s, 3H), 0.86 (s, 3H). ES(+) *m*/*z* = 370.12 [M+H].

**[5b]** *2-amino-7,8-dimethyl-4-(4-methylphenyl)-5-oxo-1-phenyl-1,4,5,6,7,8-hexahydroquinoline-3-carbonitrile* [14]. White solid. M.P. 242–244 °C. IR (KBr) (ν, cm^−1^): 3460, 3339, 3217, 3040, 2953, 2903, 2868, 2180, 1657, 1620, 1595, 1572, 1493, 1414, 1373, 1260, 1242, 1182, 1146, 1126, 1040, 851, 758, 700, 646. ^1^H NMR (500 MHz, Chloroform-*d*) δ 7.62–7.55 (m, 3H), 7.33–7.28 (m, 2H), 7.24 (d, *J* = 8.2 Hz, 2H), 7.12 (d, *J* = 7.9 Hz, 2H), 4.72 (s, 1H), 3.99 (s, 2H), 2.31 (s, 3H), 2.20 (d, *J* = 16.4 Hz, 1H), 2.14 (dd, *J* = 16.3, 1.3 Hz, 1H), 2.04 (dd, *J* = 17.4, 1.2 Hz, 1H), 1.81 (dd, *J* = 17.4, 1.3 Hz, 1H), 0.96 (s, 3H), 0.85 (s, 3H). ^13^C NMR (126 MHz, Chloroform-*d*) δ 195.67, 150.15, 149.17, 142.77, 136.36, 129.79, 129.28, 126.99, 120.98, 113.32, 63.67, 41.71, 35.64, 32.40, 29.45, 21.11.

**[5c]** *2-amino-4-(4-chlorophenyl)-7,8-dimethyl-5-oxo-1-phenyl-1,4,5,6,7,8-hexahydroquinoline-3-carbonitrile* [14]. White solid. M.P. 268–270 °C. IR (KBr) (ν, cm^−1^): 3329, 3217, 3022, 2953, 2903, 2868, 2180, 1653, 1595, 1570, 1491, 1418, 1260, 1146, 1042, 1013, 845, 698. ^1^H NMR (500 MHz, Chloroform-*d*) δ 7.67–7.57 (m, 3H), 7.35–7.30 (m, 6H), 4.77 (s, 1H), 4.09 (s, 2H), 2.23 (d, *J* = 16.4 Hz, 1H), 2.16 (dd, *J* = 16.4, 1.3 Hz, 1H), 2.06 (dd, *J* = 17.5, 1.1 Hz, 1H), 1.82 (dd, *J* = 17.4, 1.3 Hz, 1H), 0.98 (s, 3H), 0.85 (s, 3H).

**[5d]** *2-amino-4-(2,4-dichlorophenyl)-7,8-dimethyl-5-oxo-1-phenyl-1,4,5,6,7,8-hexahydroquinoline-3-carbonitrile* [14]. White solid. M.P. 256–258 °C. IR (KBr) (ν, cm^−1^): 3445, 3348, 3221, 3059, 2957, 2176, 1649, 1562, 1493, 1422, 1371, 1256, 1155, 1049, 864, 845, 723, 706. ^1^H NMR (500 MHz, Chloroform-*d*) δ 7.64–7.55 (m, 3H), 7.36 (d, *J* = 2.2 Hz, 1H), 7.34–7.28 (m, 3H), 7.21 (dd, *J* = 8.3, 2.2 Hz, 1H), 5.13 (s, 1H), 4.00 (s, 2H), 2.17 (d, *J* = 16.4 Hz, 1H), 2.11 (dd, *J* = 16.3, 1.3 Hz, 1H), 2.01 (dd, *J* = 17.5, 1.4 Hz, 1H), 1.83 (dd, *J* = 17.4, 1.3 Hz, 1H), 0.95 (s, 3H), 0.89 (s, 3H).

**[5e]** *2-amino-4-(4-fluorophenyl)-7,8-dimethyl-5-oxo-1-phenyl-1,4,5,6,7,8-hexahydroquinoline-3-carbonitrile* [14]. White solid. M.P. 258–260 °C. IR (KBr) (ν, cm^−1^): 3333, 3217, 3182, 3067, 3034, 2959, 2891, 2870, 2181, 1655, 1595, 1493, 1416, 1375, 1317, 1259, 1219, 1155, 1043, 856, 762, 704. ^1^H NMR (500 MHz, Chloroform-*d*) δ 7.65–7.59 (m, 3H), 7.39–7.30 (m, 4H), 7.06–6.96 (m, 2H), 4.78 (s, 1H), 4.20–3.93 (m, 2H), 2.23 (d, *J* = 16.4 Hz, 1H), 2.16 (dd, *J* = 16.3, 1.3 Hz, 1H), 2.05 (dd, *J* = 17.4, 1.3 Hz, 1H), 1.82 (dd, *J* = 17.4, 1.3 Hz, 1H), 0.98 (s, 3H), 0.85 (s, 3H).

**[5f]** *2-amino-7,8-dimethyl-5-oxo-1-phenyl-4-thiophen-2-yl-1,4,5,6,7,8-hexahydroquinoline-3-carbonitrile* [14]. White solid. M.P. 204–206 °C. IR (KBr) (ν, cm^−1^): 3321, 3213, 3059, 2959, 2932, 2870, 2180, 1641, 1595, 1564, 1493, 1414, 1375, 1260, 1144, 1040, 775, 760, 691. ^1^H NMR (500 MHz, Chloroform-*d*) δ 7.66–7.55 (m, 3H), 7.38–7.31 (m, 2H), 7.14 (dd, *J* = 5.1, 1.2 Hz, 1H), 7.04 (dd, *J* = 3.4, 1.0 Hz, 1H), 6.95 (dd, *J* = 5.1, 3.5 Hz, 1H), 5.13 (s, 1H), 4.12 (s, 2H), 2.27 (d, *J* = 16.3 Hz, 1H), 2.23 (dd, *J* = 16.2, 1.1 Hz, 1H), 2.08 (d, *J* = 17.5 Hz, 1H), 1.79 (dd, *J* = 17.4, 1.1 Hz, 1H), 0.99 (s, 3H), 0.89 (s, 3H).

**[5g]** *2-amino-7,8-dimethyl-5-oxo-1-phenyl-4-(3-nitrophenyl)-1,4,5,6,7,8-hexahydroquinoline-3-carbonitrile* [14]. Yellow solid. M.P. 254–255 °C. IR (KBr) (ν, cm^−1^): 3466, 3329, 3217, 3067, 2961, 2872, 2181, 1653, 1598, 1568, 1524, 1418, 1375, 1350, 1258, 1148, 1045, 926, 826, 812, 737, 694. ^1^H NMR (500 MHz, DMSO-*d*_6_) δ 8.20–8.02 (m, 2H), 7.79 (d, *J* = 7.6 Hz, 1H), 7.67 (t, *J* = 8.3 Hz, 1H), 7.65–7.56 (m, 3H), 7.50–7.36 (m, 2H), 5.52 (s, 2H), 4.66 (s, 1H), 2.25 (d, *J* = 8.6 Hz, 1H), 2.21 (d, *J* = 8.6 Hz, 1H), 2.03 (d, *J* = 16.1 Hz, 1H), 1.74 (d, *J* = 17.5 Hz, 1H), 0.89 (s, 3H), 0.73 (s, 3H).

**[5h]** *2-amino-4-(4-methoxyphenyl)-7,8-dimethyl-5-oxo-1-phenyl-1,4,5,6,7,8-hexahydroquinoline-3-carbonitrile* [14]. White solid. M.P. 230–232 °C. IR (KBr) (ν, cm^−1^): 3460, 3331, 3219, 3067, 2992, 2957, 2940, 2897, 2872, 2837, 2178, 1641, 1510, 1414, 1375, 1256, 1240, 1179, 1144, 1036, 851, 818, 762, 706, 665. ^1^H NMR (500 MHz, Chloroform-*d*) δ 7.64–7.54 (m, 3H), 7.36–7.26 (m, 5H), 6.85 (d, *J* = 8.7 Hz, 2H), 4.71 (s, 1H), 3.98 (s, 2H), 3.78 (s, 3H), 2.20 (d, *J* = 16.3 Hz, 1H), 2.14 (dd, *J* = 16.3, 1.3 Hz, 1H), 2.03 (dd, *J* = 17.4, 1.3 Hz, 1H), 1.80 (dd, *J* = 17.4, 1.3 Hz, 1H), 0.95 (s, 3H), 0.84 (s, 3H).

**[5i]** *Methyl 2-amino-4-(2,4-dichlorophenyl)-7,8-dimethyl-5-oxo-1-phenyl-1,4,5,6,7,8-hexahydroquinoline-3-carboxylate*. Novel. White solid. M.P. 213–215 °C. IR (KBr) (ν, cm^−1^): 3437, 3364, 3256, 3198, 3057, 2955, 2870, 1665, 1595, 1491, 1439, 1373, 1275, 1213, 1180, 1045, 847, 816, 785, 760, 710, 625. ^1^H NMR (500 MHz, Chloroform-*d*) δ 7.68–7.56 (m, 3H), 7.46 (d, *J* = 8.3 Hz, 1H), 7.40–7.33 (m, 2H), 7.30 (d, *J* = 2.2 Hz, 1H), 7.17 (dd, *J* = 8.3, 2.2 Hz, 1H), 6.27 (s, 2H), 5.35 (s, 1H), 3.59 (s, 3H), 2.18 (d, *J* = 16.3 Hz, 1H), 2.10 (dd, *J* = 16.3, 1.5 Hz, 1H), 2.02 (dd, *J* = 17.4, 1.4 Hz, 1H), 1.79 (dd, *J* = 17.4, 1.4 Hz, 1H), 0.95 (s, 3H), 0.84 (s, 3H). ^13^C NMR (126 MHz, Chloroform-*d*) δ 195.71, 170.32, 152.11, 149.95, 142.96, 136.22, 134.05, 133.41, 131.64, 130.53, 130.24, 130.19, 129.46, 126.32, 112.92, 79.03, 50.46, 49.99, 42.05, 34.77, 32.28, 30.93, 29.57, 26.84. ES(+) *m*/*z* = 471.30–475.27 ([M+H], isotopes).

**[5j]** *2-amino-7,8-dimethyl-5-oxo-1-(3-methylphenyl)-4-(3-nitrophenyl)-1,4,5,6,7,8-hexahydroquinoline-3-carbonitrile*. Novel yellow solid. M.P. 246–248 °C.IR (KBr) (ν, cm^−1^): 3329, 3217, 3034, 2961, 2932, 2872, 2181, 1653, 1568, 1524, 1418, 1375, 1350, 1260, 1179, 1152, 1092, 1047, 789, 743, 694. ^1^H NMR (500 MHz, DMSO-*d*_6_) δ 8.19–8.03 (m, 2H), 7.78 (d, *J* = 7.6 Hz, 1H), 7.67 (t, *J* = 7.8 Hz, 1H), 7.50 (t, *J* = 7.7 Hz, 1H), 7.40 (d, *J* = 7.7 Hz, 1H), 7.32–7.12 (m, 2H), 5.51 (s, 2H), 4.65 (s, 1H), 2.41 (s, 3H), 2.23 (dd, *J* = 16.8, 11.3 Hz, 2H), 2.03 (d, *J* = 16.8 Hz, 1H), 1.79 (d, *J* = 17.5 Hz, 1H), 0.90 (s, 3H), 0.73 (s, 3H). ^13^C NMR (126 MHz, DMSO-*d*_6_) δ 195.32, 152.09, 151.49, 149.17, 148.36, 140.57, 136.34, 134.20, 131.03, 130.63, 130.46, 127.25, 121.97, 121.62, 111.50, 59.66, 49.67, 41.38, 36.89, 32.45, 29.44, 26.72, 21.36. ES(+) *m*/*z* = 429.34 [M+H].

**[5k]** *2-amino-1-(3-chlorophenyl)-4-(2,4-dichlorophenyl)-7,8-dimethyl-5-oxo-1,4,5,6,7,8-hexahydroquinoline-3-carbonitrile*. Novel white solid. M.P. 240–242 °C. IR (KBr) (ν, cm^−1^): 3466, 3343, 3217, 3067, 2955, 2901, 2870, 2183, 1657, 1618, 1589, 1570, 1468, 1373, 1260, 1146, 1103, 1047, 860, 795, 718, 702. ^1^H NMR (500 MHz, DMSO-*d*_6_) δ 7.70–7.58 (m, 3H), 7.52 (d, *J* = 2.1 Hz, 1H), 7.48–7.36 (m, 3H), 5.51 (s, 2H), 4.98 (s, 1H), 2.28–2.11 (m, 2H), 1.97 (d, *J* = 16.0 Hz, 1H), 1.78 (d, *J* = 17.4 Hz, 1H), 0.91 (s, 3H), 0.81 (s, 3H). ^13^C NMR (126 MHz, DMSO-*d*_6_) δ 195.14, 151.60, 151.45, 143.51, 137.73, 134.66, 132.92, 131.98, 131.88, 131.76, 130.92, 130.60, 129.60, 128.90, 128.28, 121.23, 110.94, 59.51, 49.66, 41.48, 34.12, 32.41, 29.49, 26.99. ES(+) *m*/*z* = 471.30–477.23 ([M+H]^+^, isotopes).

**[5l]** *2-amino-4-(2,4-dichlorophenyl)-7,8-dimethyl-1-(3-nitrophenyl)-5-oxo-1,4,5,6,7,8-hexahydroquinoline-3-carbonitrile*. Novel yellow solid. M.P. 252–254 °C. IR (KBr) (ν, cm^−1^): 3377, 3331, 3227, 3096, 3069, 2972, 2955, 2870, 2189, 1661, 1649. ^1^H NMR (500 MHz, DMSO-*d*_6_) δ 8.41 (ddd, *J* = 8.2, 2.3, 1.3 Hz, 1H), 8.36 (t, *J* = 2.3 Hz, 1H), 7.92 (dt, *J* = 8.2, 1.3 Hz, 1H), 7.86 (t, *J* = 8.2 Hz, 1H), 7.53 (d, *J* = 2.2 Hz, 1H), 7.48 (d, *J* = 8.4 Hz, 1H), 7.40 (dd, *J* = 8.4, 2.2 Hz, 1H), 5.64 (s, 2H), 5.08–4.87 (m, 1H), 2.27–2.13 (m, 2H), 1.97 (d, *J* = 16.0 Hz, 1H), 1.79 (d, *J* = 17.4 Hz, 1H), 0.89 (s, 3H), 0.85 (s, 3H). ^13^C NMR (126 MHz, DMSO-*d*_6_) δ 195.24, 151.60, 151.38, 149.29, 143.46, 137.48, 137.41, 133.00, 132.01, 131.78 (d, *J* = 1.9 Hz), 128.91, 128.22, 126.49, 125.28, 121.21, 111.01, 59.59, 56.51, 49.67, 41.57, 34.28, 32.44, 29.48, 26.93, 19.01. ES(+) *m*/*z* = 483.34–487.31 ([M+H], isotopes).

**[5m]** *2-amino-1-(4-methoxyphenyl)-7,8-dimethyl-4-(3-methylphenyl)-5-oxo-1,4,5,6,7,8-hexahydroquinoline-3-carbonitrile*. Novel white solid. M.P. 172–174 °C. IR (KBr) (ν, cm^−1^): 3460, 3391, 3331, 3219, 3042, 3005, 2963, 2872, 2893, 2180, 1661, 1618, 1566, 1514, 1414, 1375, 1294, 1256, 1171, 1146, 1042, 1024, 851, 810, 779, 708. ^1^H NMR (500 MHz, Chloroform-*d*) δ 7.26–7.16 (m, 4H), 7.14 (d, *J* = 7.5 Hz, 1H), 7.08 (d, *J* = 8.3 Hz, 2H), 7.02 (d, *J* = 7.4 Hz, 1H), 4.72 (s, 1H), 4.06 (s, 2H), 3.92 (s, 3H), 2.37 (s, 3H), 2.26–2.14 (m, 2H), 2.07 (d, *J* = 17.5 Hz, 1H), 1.88 (dd, *J* = 17.4, 1.3 Hz, 1H), 1.00 (s, 3H), 0.89 (s, 3H). ^13^C NMR (126 MHz, Chloroform-*d*) δ 195.65, 160.56, 150.47, 149.77, 145.64, 137.91, 130.78, 128.47, 128.44, 128.03, 127.44, 123.96, 121.06, 115.66, 113.19, 63.51, 58.45, 55.70, 49.98, 41.70, 35.91, 32.37, 29.50, 27.06, 21.65, 18.43. ES(+) *m*/*z* = 414.16 [M+H].

**[5n]** *2-amino-4-(2,4-dichlorophenyl)-7,8-dimethyl-1-(methylphenyl)-5-oxo-1,4,5,6,7,8-hexahydroquinoline-3-carbonitrile*. Novel white solid. M.P. 229–231 °C. IR (KBr) (ν, cm^−1^): 3472, 3333, 3237, 3067, 2959, 2870, 2183, 1653, 1628, 1570, 1474, 1425, 1379, 1240, 1175, 1049, 970, 943, 851, 835, 696. ^1^H NMR (500 MHz, Chloroform-*d*) δ 7.56–7.46 (m, 2H), 7.43 (t, *J* = 7.4 Hz, 1H), 7.37 (d, *J* = 2.0 Hz, 1H), 7.28–7.25 (m, 2H), 7.18–7.09 (m, 2H), 5.19 (s, 1H), 5.04 (d, *J* = 18.1 Hz, 1H), 4.93 (d, *J* = 18.1 Hz, 1H), 4.04 (s, 2H), 2.60 (d, *J* = 16.5 Hz, 1H), 2.33–2.15 (m, 4H), 1.09 (s, 3H), 1.05 (s, 3H). ^13^C NMR (126 MHz, Chloroform-*d*) δ 195.14, 151.88, 151.24, 140.79, 135.81, 133.67, 132.91, 131.04, 129.87, 129.85, 128.68, 127.28, 125.05, 120.27, 112.73, 65.38, 49.80, 48.47, 40.27, 34.68, 33.09, 29.18, 27.36. ES(+) *m*/*z* = 452.08–457.25 ([M+H], isotopes).

**[5o]** *2-amino-1-cyclohexyl-7,8-dimethyl-5-oxo-4-(3-nitrophenyl)-1,4,5,6,7,8-hexahydroquinoline-3-carbonitrile*. Traces collected of novel yellow solid. ^1^H NMR (500 MHz, Chloroform-*d*) δ 8.03 (dd, *J* = 8.1, 2.4 Hz, 1H), 7.98 (t, *J* = 2.1 Hz, 1H), 7.70 (d, *J* = 7.9 Hz, 1H), 7.44 (td, *J* = 7.9, 1.8 Hz, 1H), 4.82 (d, *J* = 2.6 Hz, 1H), 4.46 (d, *J* = 7.2 Hz, 2H), 3.72 (tt, *J* = 12.6, 3.5 Hz, 1H), 2.73 (d, *J* = 16.3 Hz, 1H), 2.35–2.20 (m, 4H), 2.03–1.86 (m, 3H), 1.84–1.66 (m, 4H), 1.45–1.29 (m, 2H), 1.23–0.95 (m, 8H).

**[5p]** *2-amino-4-(4-cyanophenyl)-7,8-dimethyl-1-(4-(phenylazo)phenyl)-5-oxo-1,4,5,6,7,8-hexahydroquinoline-3-carbonitrile*. Novel orange solid. M.P. 238–240 °C. IR (KBr) (ν, cm^−1^): 3337, 3057, 2959, 2872, 2230, 2183, 1630, 1499, 1414, 1371, 1315, 1260, 1153, 1040, 868, 843, 777, 692, 584, 569. ^1^H NMR (500 MHz, Chloroform-*d*) δ 8.21–8.09 (m, 2H), 8.07–7.95 (m, 2H), 7.75–7.63 (m, 2H), 7.63–7.55 (m, 3H), 7.55–7.42 (m, 4H), 4.86 (s, 1H), 4.19 (s, 2H), 2.26 (d, *J* = 16.4 Hz, 1H), 2.22–2.08 (m, 2H), 1.93 (d, *J* = 17.6 Hz, 1H), 1.01 (s, 3H), 0.87 (s, 3H). ^13^C NMR (126 MHz, DMSO-*d*_6_) δ 195.31, 152.53, 152.44, 152.34, 151.84, 151.12, 138.88, 132.99, 132.56, 131.95, 130.10, 128.52, 124.53, 123.19, 121.60, 119.43, 111.29, 109.64, 59.72, 49.72, 41.52, 37.56, 32.45, 29.45, 26.81. ES(+) *m*/*z* = 499.41 [M+H].

**[5q]** *2-amino-4-(2,4-dichlorophenyl)-7,8-dimethyl-1-(4-(3-methylphenylazo)-3-methylphenyl)-5-oxo-1,4,5,6,7,8-hexahydroquinoline-3-carbonitrile.* Novel orange solid. M.P. 255–257 °C. IR (KBr) (ν, cm^−1^): 3489, 3393, 3157, 3126, 3084, 3063, 2957, 2870, 2179, 1734, 1653, 1603, 1568, 1470, 1418, 1373, 1261, 1176, 1109, 1047, 887, 864, 851, 793, 690, 669. ^1^H NMR (500 MHz, Chloroform-*d*) δ 7.81–7.72 (m, 3H), 7.45 (dd, *J* = 8.5, 7.5 Hz, 1H), 7.40–7.34 (m, 2H), 7.32 (d, *J* = 8.5 Hz, 1H), 7.27 (d, *J* = 2.2 Hz, 1H), 7.25–7.18 (m, 2H), 5.14 (s, 1H), 4.07 (d, *J* = 14.0 Hz, 2H), 2.81 (s, 3H), 2.49 (s, 3H), 2.19 (d, *J* = 16.4 Hz, 1H), 2.14 (d, *J* = 1.2 Hz, 1H), 2.12–2.05 (m, 1H), 1.94 (d, *J* = 17.4 Hz, 1H), 0.98 (s, 3H), 0.92 (s, 3H). ^13^C NMR (126 MHz, Chloroform-*d*) δ 195.45, 152.82, 151.61, 150.30, 150.25, 140.74, 140.10, 139.30, 137.35, 133.83, 132.98, 132.80, 132.41, 131.70, 130.05, 129.14, 128.09, 127.28, 123.69, 120.71, 120.31, 117.83, 111.11, 61.99, 49.84, 41.82, 35.49, 32.43, 29.39, 27.24, 21.42, 17.74. ES(+) *m*/*z* = 570.35–574.34 ([M+H], isotopes).

**[5s]** *2-amino-4-(4-methylphenyl)-5-oxo-1-phenyl-1,4,5,6,7,8-hexahydroquinoline-3-carbonitrile* [14]. Brownish solid. M.P. 235–237 °C. IR (KBr) (ν, cm^−1^): 3314, 3211, 3063, 3044, 3013, 2953, 2930, 2889, 2866, 2183, 1638, 1618, 1595, 1568, 1512, 1495, 1456, 1414, 1371, 1310, 1267, 1192, 1136, 1080, 1005, 901, 837, 829, 760, 698, 550. ^1^H NMR (500 MHz, Chloroform-*d*) δ 7.64–7.54 (m, 3H), 7.38–7.31 (m, 2H), 7.28 (d, *J* = 7.3 Hz, 3H), 7.15 (d, *J* = 8.0 Hz, 2H), 4.80 (s, 1H), 4.01 (d, *J* = 14.3 Hz, 2H), 2.43–2.36 (m, 1H), 2.34 (s, 3H), 2.29 (ddd, *J* = 16.8, 12.1, 4.9 Hz, 1H), 2.24–2.15 (m, 1H), 2.06–1.98 (m, 1H), 1.92 (dp, *J* = 14.4, 4.8 Hz, 1H), 1.80 (dddt, *J* = 13.4, 12.1, 10.5, 4.7 Hz, 1H).

**[5t]** *2-amino-4-(4-chlorophenyl)-5-oxo-1-phenyl-1,4,5,6,7,8-hexahydroquinoline-3-carbonitrile* [14]. Yellow solid. M.P. 228–230 °C. IR (KBr) (ν, cm^−1^): 3318, 3215, 3063, 3044, 2972, 2949, 2893, 2185, 1638, 1618, 1595, 1566, 1491, 1414, 1373, 1269, 1246, 1192, 1138, 1090, 1016, 105, 903, 833, 760, 700, 637. ^1^H NMR (500 MHz, Chloroform-*d*) δ 7.66–7.55 (m, 3H), 7.37–7.28 (m, 6H), 4.82 (s, 1H), 4.06 (s, 2H), 2.39 (dt, *J* = 16.6, 4.8 Hz, 1H), 2.30 (ddd, *J* = 16.8, 12.0, 4.8 Hz, 1H), 2.25–2.14 (m, 1H), 2.07–1.98 (m, 1H), 1.93 (dp, *J* = 14.4, 4.8 Hz, 1H), 1.85–1.73 (m, 1H).

**[7a]** *2′-Amino-7‚7′-dimethyl-2,5′-dioxo-1′-phenyl-5′‚6′‚7′‚8′-tetrahydro-1H-spiro[indoline-3,4′-quinoline]-3′-carbonitrile* [21]. White solid. M.P. 304–306 °C. IR (KBr) (ν, cm^−1^): 3431, 3325, 3217, 3055, 3092, 3055, 2965, 2872, 2191, 1721, 1699, 1618, 1645, 1595, 1558, 1472, 1422, 1364, 1341, 1317, 1263, 1198, 103, 926, 743, 729, 689. ^1^H NMR (500 MHz, DMSO-*d*_6_) δ 10.21 (s, 1H), 7.67–7.56 (m, 3H), 7.48 (s, 2H), 7.18 (d, *J* = 7.3 Hz, 1H), 7.13 (td, *J* = 7.6, 1.3 Hz, 1H), 6.92 (td, *J* = 7.5, 1.0 Hz, 1H), 6.77 (d, *J* = 7.6 Hz, 1H), 5.35 (s, 2H), 2.12 (dd, *J* = 16.6, 13.8 Hz, 2H), 1.94 (d, *J* = 15.9 Hz, 1H), 1.82 (d, *J* = 17.3 Hz, 1H), 0.89 (s, 3H), 0.82 (s, 3H). ^13^C NMR (126 MHz, DMSO-*d*_6_) δ 194.36, 179.93, 152.35, 151.58, 141.93, 137.14, 136.47, 130.78, 130.42, 128.15, 123.64, 121.86, 119.39, 110.88, 109.31, 61.42, 56.51, 49.79, 48.98, 41.85, 32.59, 28.70, 27.10, 19.04. ES(+) *m*/*z* = 411.13 [M+H].

**[7b]** *2′-Amino-7′‚7′-dimethyl-1′-(3-methylphenyl)-2,5′-dioxo-5′‚6′‚7′‚8′-tetrahydro-1H-spiro[indoline-3,4′-quinoline]-3′-carbonitrile.* Novel white solid. MPT: 304–306 °C. IR (KBr) (ν, cm^−1^): 3460, 3356, 3333, 3051, 2959, 2870, 2183, 1715, 1692, 1645, 1558, 1474, 1364, 1341, 1315, 1261, 1215, 1051, 743, 691. ^1^H NMR (500 MHz, DMSO-*d*_6_) δ 10.21 (s, 1H), 7.50 (t, *J* = 7.7 Hz, 1H), 7.44 (d, *J* = 7.6 Hz, 1H), 7.37–7.19 (m, 2H), 7.17 (d, *J* = 7.1 Hz, 1H), 7.14 (t, *J* = 7.4 Hz, 1H), 6.92 (t, *J* = 7.4 Hz, 1H), 6.77 (d, *J* = 7.6 Hz, 1H), 5.34 (s, 2H), 2.42 (s, 3H), 2.23–2.03 (m, 2H), 1.94 (d, *J* = 16.0 Hz, 1H), 1.86 (d, *J* = 17.2 Hz, 1H), 0.89 (s, 3H), 0.83 (s, 3H). ^13^C NMR (126 MHz, DMSO-*d*_6_) δ 194.35, 179.95, 152.40, 151.57, 141.93, 140.62, 137.17, 136.31, 131.07, 130.48, 128.13, 127.31 (d, *J* = 50.2 Hz), 123.64, 121.86, 119.42, 110.81, 109.30, 61.29, 49.82, 48.97, 41.78, 32.61, 28.68, 27.15, 21.31. ES(+) *m*/*z* = 426.35 [M+H].

**[7c]** *2′-Amino-7′‚7′-dimethyl-1′-(3-nitrophenyl)-2,5′-dioxo-5′‚6′‚7′‚8′-tetrahydro-1H-spiro[indoline-3,4′-quinoline]-3′-carbonitrile* [21]. Light yellow solid. M.P. 324–326 °C. IR (KBr) (ν, cm^−1^): 3395, 3333, 3298, 3096, 2959, 2874, 2187, 1746, 1649, 1543, 1472, 1366, 1315, 1258, 1225, 1198, 1153, 1049, 926, 762, 725, 696, 675, 619, 579. ^1^H NMR (500 MHz, DMSO-*d*_6_) δν, cm^−1^): 3460, 3325, 3215, 3069, 1957, 1945, 2905, 2872, 2189, 1734, 1719, 1653, 1514, 1472, 1364, 1252, 1190. 1–49, 1028, 920, 851, 762, 739, 691. ^1^H NMR (500 MHz, DMSO-*d*_6_) δ 10.19 (s, 1H), 7.43 (d, *J* = 6.2 Hz, 1H), 7.35 (d, *J* = 8.4 Hz, 1H), 7.23–7.06 (m, 4H), 6.91 (td, *J* = 7.5, 1.0 Hz, 1H), 6.76 (d, *J* = 7.6 Hz, 1H), 5.36 (s, 2H), 3.86 (s, 3H), 2.14 (d, *J* = 17.3 Hz, 1H), 2.10 (d, *J* = 16.0 Hz, 1H), 1.93 (d, *J* = 17.0 Hz, 1H), 1.87 (dd, *J* = 17.3, 1.3 Hz, 1H), 0.90 (s, 3H), 0.83 (s, 3H).

**[7e]** *2′-Amino-1′-(4-chlorophenyl)-7′‚7′-dimethyl-2,5′-dioxo-5′‚6′‚7′‚8′-tetrahydro-1H-spiro[indoline-3,4′-quinoline]-3′-carbonitrile* [21]. White solid. M.P. 298–300 °C. IR (KBr) (ν, cm^−1^): 3420, 3329, 294, 3055, 2963, 2870, 2191, 1749, 1566, 1641, 1618, 1491, 1472, 1416, 1364, 1315, 1263, 1194, 1092, 1051, 1018, 918, 799, 756, 745, 689. ^1^H NMR (500 MHz, DMSO-*d*_6_) δ 10.20 (s, 1H), 7.73–7.62 (m, 2H), 7.52 (s, 2H), 7.18 (dd, *J* = 7.3, 1.2 Hz, 1H), 7.12 (td, *J* = 7.6, 1.3 Hz, 1H), 6.91 (td, *J* = 7.4, 1.0 Hz, 1H), 6.76 (d, *J* = 7.6 Hz, 1H), 5.55 (s, 2H), 2.15 (d, *J* = 17.3 Hz, 1H), 2.10 (d, *J* = 15.9 Hz, 1H), 1.93 (dd, *J* = 15.9, 1.2 Hz, 1H), 1.83 (dd, *J* = 17.3, 1.3 Hz, 1H), 0.90 (s, 3H), 0.83 (s, 3H).

**[7f]** *2′-Amino-1′-(4-bromophenyl)-7′‚7′-dimethyl-2,5′-dioxo-5′‚6′‚7′‚8′-tetrahydro-1H-spiro[indoline-3,4′-quinoline]-3′-carbonitrile* [21]. White solid. M.P. 300–302 °C. IR (KBr) (ν, cm^−1^): 3418, 3321, 3291, 3057, 2959, 2868, 2193, 1749, 1661, 1649, 1618, 1564, 1487, 1474, 1364, 1315, 1263, 1196, 1150, 1051, 1015, 916, 797, 754, 745, 689, 637, 623. ^1^H NMR (500 MHz, DMSO-*d*_6_) δ 10.20 (s, 1H), 7.80 (d, *J* = 7.5 Hz, 2H), 7.45 (s, 2H), 7.18 (dd, *J* = 7.3, 1.2 Hz, 1H), 7.12 (td, *J* = 7.6, 1.3 Hz, 1H), 6.91 (td, *J* = 7.5, 1.0 Hz, 1H), 6.76 (d, *J* = 7.6 Hz, 1H), 5.54 (s, 2H), 2.14 (d, *J* = 17.3 Hz, 1H), 2.10 (d, *J* = 16.0 Hz, 1H), 1.95 (d, *J* = 16.0 Hz, 1H), 1.83 (dd, *J* = 17.3, 1.2 Hz, 1H), 0.90 (s, 3H), 0.83 (s, 3H).

**[7g]** *2′-Amino-7′‚7′-dimethyl-1′-(4-methylphenyl)-2,5′-dioxo-5′‚6′‚7′‚8′-tetrahydro-1H-spiro[indoline-3,4′-quinoline]-3′-carbonitrile* [21]. White solid. M.P. 298–300 °C. IR (KBr) (ν, cm^−1^): 3435, 3321, 3215, 3092, 3059, 3024, 2959, 2870, 2189, 1722, 1697, 1645, 1558, 1508, 1474, 1364, 1341, 1315, 1261, 1198, 1152, 1051, 922, 752, 689, 627. ^1^H NMR (500 MHz, DMSO-*d*_6_) δ 10.20 (s, 1H), 7.42 (d, *J* = 7.8 Hz, 2H), 7.36 (s, 2H), 7.21–7.14 (m, 1H), 7.12 (td, *J* = 7.6, 1.3 Hz, 1H), 6.91 (td, *J* = 7.5, 1.0 Hz, 1H), 6.76 (d, *J* = 7.6 Hz, 1H), 5.32 (s, 2H), 2.42 (s, 3H), 2.12 (dd, *J* = 18.7, 16.6 Hz, 2H), 1.93 (d, *J* = 15.8 Hz, 1H), 1.85 (d, *J* = 17.2 Hz, 1H), 0.89 (s, 3H), 0.82 (s, 3H).

**[7i]** *2′-Amino-1′-(3-chlorophenyl)-7′‚7′-dimethyl-2,5′-dioxo-5′‚6′‚7′‚8′-tetrahydro-1H-spiro[indoline-3,4′-quinoline]-3′-carbonitrile*. Novel white solid. M.P. 314–316 °C. IR (KBr) (ν, cm^−1^): 3437, 3325, 3264, 3078, 3051, 3017, 2955, 2870, 2191, 1749, 1653, 1589, 1568, 1472, 1418, 1364, 1317, 1265, 1196, 1194, 1152, 1096, 1051, 922, 887, 849, 810, 750, 718, 689, 677, 613. ^1^H NMR (500 MHz, DMSO-*d*_6_) δ 10.20 (s, 1H), 7.85–7.55 (m, 3H), 7.46 (s, 1H), 7.24 (s, 1H), 7.12 (td, *J* = 7.7, 1.3 Hz, 1H), 6.91 (td, *J* = 7.5, 1.0 Hz, 1H), 6.76 (d, *J* = 7.5 Hz, 1H), 5.55 (s, 2H), 2.25–2.02 (m, 2H), 1.98–1.89 (m, 1H), 1.83 (d, *J* = 17.3 Hz, 1H), 0.90 (s, 3H), 0.83 (s, 3H). ^13^C NMR (126 MHz, DMSO-*d*_6_) δ 194.40, 179.91, 152.03, 151.52, 141.92, 137.84, 137.07, 134.75, 132.06, 130.82, 130.65, 129.49, 128.15, 123.87, 121.84, 119.37, 111.02, 109.27, 61.47, 49.83, 49.00, 41.78, 32.63, 28.72, 27.12. ES(+) *m*/*z* = 448.24 [M+H].

**[12]** *2-amino-1-[4-[(4-bromophenyl)(1H-indol-3-yl)methyl]phenyl]-7,8-dimethyl-5-oxo-4-(3-methylphenyl)-1,4,5,6,7,8-hexahydroquinoline-3-carbonitrile*. Novel yellow solid. M.P. 170–174 °C. IR (KBr) (ν, cm^−1^): 3377, 3355, 3219, 3055, 2957, 2868, 2181, 1636, 1562, 1485, 1456,1408, 1371, 1258, 1146, 1103, 1042, 1009, 837, 743, 702, 615. ^1^H NMR (500 MHz, Chloroform-*d*) δ 8.12 (s, 1H), 7.48 (d, *J* = 8.5 Hz, 2H), 7.45–7.39 (m, 3H), 7.28–7.22 (m, 4H), 7.21–7.19 (m, 3H), 7.13 (d, *J* = 8.4 Hz, 2H), 7.09 (dt, *J* = 7.8, 1.8 Hz, 1H), 7.03 (ddt, *J* = 8.0, 6.9, 1.0 Hz, 1H), 6.68 (d, *J* = 1.2 Hz, 1H), 5.75 (s, 1H), 4.74 (s, 1H), 4.10 (d, *J* = 13.9 Hz, 2H), 2.35 (s, 3H), 2.25–2.14 (m, 2H), 2.06 (dd, *J* = 17.8, 7.5 Hz, 1H), 1.84 (d, *J* = 17.4, 1H), 0.99 (d, *J* = 6.3 Hz, 3H), 0.86 (d, *J* = 6.9 Hz, 3H). ^13^C NMR (126 MHz, DMSO-*d*_6_) δ 195.71, 150.52/150.50, 149.75/149.74, 147.01, 144.73, 138.28, 136.84, 133.89/133.86, 131.62, 131.16, 129.68, 129.42, 129.00, 128.99, 128.55, 127.66, 126.70/126.69, 125.96, 124.07/124.06, 122.38, 120.86, 120.47, 119.54/119.53, 119.45/119.42, 118.76/118.71, 112.84, 111.46, 62.61/62.59, 49.96/49.60, 47.71/47.70, 35.76, 32.40, 29.45/29.44, 27.03, 21.57. ES(+) *m*/*z* = 669.51 [M+H].

## 4. Conclusions

A wide range (27 examples, 12 of which novel) of *N*-aryl-4-aryldihydropyridines were synthesized using a novel, cheap, mild, and metal-free biopolymer-based catalyst, pip–agar, via the combination of aldehydes or isatin, anilines, dimedone, and malononitrile or methyl cyanoacetate. Table 5 showcases the advantages of our developed system in comparison to previous studies (excluding the studies involving the two-component version of the reaction). In addition, in an innovative and unprecedented approach, one of the products of the aza-Friedel–Crafts multicomponent reaction, catalysed by a heterogeneous catalyst previously developed by the same authors (silicotungstic acid supported on Amberlyst^®^ 15), was used instead of aniline in dihydropyridine synthesis to create a complex novel scaffold.

## Data Availability

The data presented in this study are available in the article or Appendix A.

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
