# Peer review of "Novel Biopolymer-Based Catalyst for the Multicomponent Synthesis of N-aryl-4-aryl-Substituted Dihydropyridines Derived from Simple and Complex Anilines"

_molecules, 2024, doi:10.3390/molecules29081884_

Round 1
Reviewer 1 Report
Comments and Suggestions for Authors
The manuscript submitted by Bosica and Abdilla, entitled “Novel biopolymer-based catalyst for the multicomponent synthesis of N-aryl-4-aryl substituted dihydropyridines deriving from simple and complex anilines” describe the application of novel, biopolymer-based catalyst in multicomponent synthesis of substituted DHPs. Authors have covered application of Agar-pip catalyst in synthesis of 13 known molecules as well as 12 novel compounds. Although, the studied catalyst provided worst results than known methods (lower yields and longer reaction time), the ease of preparation and price are in favor here. Thus I think, that the manuscript is well-suited for Molecules special issue “Multicomponent Reactions in Organic Synthesis”, however few thing required clarification and correction:
1. In entire manuscript, references leading to specific figure or table are missing. Instead “Error! Reference source not found.” is displayed. Please correct this in final version of the manuscript.
2. Authors mix the method of referring to specific experiments. Sometimes it is listed as trial, sometimes as entries etc. Please chose one convention over entire manuscript.
3. In table 2, entry 4, column 3 is 4-FPhCO, should rather be 4-FPhCHO
4. Footnote for table 2 is missing note c and d, please add.
5. In my opinion table 3 present data in unclear way. I suggest to add info about substituents of used anilines into the table for clarity.
6. In section 2.3.1, authors briefly described the procedure to obtain the catalyst. The description is lacking amounts of reagents and solvents used. Please add missing information.
7. In caption for fig2 and fig3, the catalyst is referred as Piperazine-Agar-Agar, however in text authors used also Pip-agar and piperazine-agar. Please hold to one convention during manuscript
8. Authors are discussing about HSQC, HMBC and DEPT experiments. Based on the results, I assume that DEPT mean DEPT135, HSQC is 1H-13C HSQC and HMBC is 1H-13C HMBC. Please change those information cause there are different types of DEPT, HSQC and HMBC spectra possible, not only those three.
9. In section 2.6 authors are discussing in great details the NMR experiments. Thus the section is well-written, it lengthens the manuscript unnecessarily. I would suggest to move this section to the Supplement.
10. Did the authors performed any purity check on studied molecules (such as HPLC analysis or Elemental analysis). If not, please add such studies especially for novel compounds.
11. In supplementary, please correct DMSO-d5 to DMSO-d6 (figure 28 and 42)
12. Comparison between known synthesis methods of 5a and those studied in paper (time, yields etc.) would increase the scientific value of the manuscript. I suggest add those information in either discussion or conclusion.
Reviewer 2 Report
Comments and Suggestions for Authors
Bosica and Abdilla reported a novel method for the synthesis of N-aryl-4-aryldihydropyridines composite scaffolds by using metal-free biopolymer-based catalyst. The authors used their previously reported aza-Friedel Crafts multicomponent reaction products to substitute aniline for the synthesis of more complex dihydropyridine. Although the results are interesting and the synthesis method is mild, the authors need to modify certain aspects of their paper before it can be accepted for publication in Molecules.
Some minor issues to address are listed below:
1) In order to reflect the experimental results more aesthetically and directly, this paper needs to modify Table 1, Table 2 and Table 3.
2) Line 168 “(several minor spots could be observed on TLC plate during reaction monitoring)”. TLC can not be used as a basis for judgment, it should be monitored by GC-MS or other means.
3) In addition to 1,3-indandione (3b, Table 2, entry 17), the authors need to try other compounds to replace 3a. We do not see any more possibilities from 3b.
4) The authors need to do more experiments with aliphatic amines before they can reach their conclusions in line 182-184.
5) Figure 2 and Figure 3 require a clearer view. Current diagrams are not yet at the level required for publication in Molecules.
6) Line 277 “In addition, TLC analysis during the course of the reaction did not elucidate any changes in the relative intensity of the enaminone intermediate and of the final product.” TLC can not be used as a basis for judgment, it should be monitored by GC or 1H NMR or other means.
Reviewer 3 Report
Comments and Suggestions for Authors
The manuscript titled Novel biopolymer-based catalyst for the multicomponent synthesis of N-aryl-4-aryl substituted dihydropyridines deriving from simple and complex anilines by Bosica and coworker describes a synthesis of N-aryl-4-aryldihydropyridines were synthesized using a novel, cheap, mild, and metal-free biopolymer-based catalyst. In my opinion, the concept and the experimental conduction are decent, but the manuscript needs some more polish. It meets the criteria of Molecules, and I would recommend for publication after the authors addressing the following comments.
Detailed notes and comments are outlined below:
1. (General) The manuscript is filled with ‘Error! Reference source not found,’ which makes the article very hard to read.
2. (Introduction) For the first couple of sentences, the authors need to reference more articles and reviews to support their statement.
3. (Introduction) The authors introduced the abbreviation of DHPs but rarely used it in the context.
4. (Schemes) General reaction conditions should be added in the scheme. Also, no brackets are needed for the compounds (using 1 instead of (1)). Full chemical names of the products are not needed as well.
5. (Scheme 1) I think the authors can draw the mechanism or the key intermediate of the MCR somewhere in the manuscript.
6. (Scheme 2) Please clean up the structure of 7 – one of the carbonyls looks distorted.
7. (Scheme 3) Benzaldehyde should be labeled as 2a instead of 1a. This scheme can also be integrated into Scheme 2 as well.
8. (Table 1) This table should be edited and removed all the entries that were not discussed in the manuscript – such as entries 4 to 9. Or at least move them to the ESI. Also, does trial also mean entry? The authors should keep them consistent. I also think the authors should explore more on the material ratio and solvent options once the best catalyst was found.
9. (Table 1) 25 mol% loading is quite high as a catalyst – since the authors claimed this is a cheap and easy-to-recover type of catalyst, can we at least try stoichiometric amount and see its efficiency?
10. (Line 132) When talking about malononitrile being in slight excess, it should be 1.2 equiv instead of 0.2 equiv to avoid any sorts of confusion.
11. (Table 2) How the authors describe benzaldehydes are not scientifically correct – for example, for 4-MePhCHO (2b), it should be written as 4-Me-C6H4CHO. Diketone was basically limited to 3a because the only other example did not give any desired product, which I think is a huge limitation of the reaction and I courage the authors explore more on the scope.
12. (Table 2) I would suggest the authors turn Table 2 into a scheme to showcase the product structures (same as Table 3).
13. (Table 3) Isatin is popular substrate in organic synthesis and there are a lot of isatin derivatives on market, therefore I think it would be beneficial that the authors try some more isatins for the scope.
14. (Section 2.3.1) Do we have any physical characterization or quantified description of the catalyst? Such as average particle size or particle size distribution?
15. (Table 4) I think there are some layout issues here such as ‘atom economy = E-factor’ which clearly isn’t right. Atom Economy should be moved to the top line instead.
16. (Scheme 5) Compound 9 should be labeled as 10 in this scheme.
17. (Figure 8) Please remove all the impurity peaks in the NMR spectrum – they only make the data clustered. I would also suggest a ChemDraw format to elucidate the structure.
18. (Section 3.5) I think the format of MS is confusing – I think only the M+H peak should be kept in the characterization. Also, melting point should be a range. For compound 12, why did this compound have 1H-NMR in chloroform but 13C-NMR in DMSO?
19. (General) The manuscript’s general format and language should be polished, e.g.:
· (Line 29) Reduction in reaction time, purification length, and solvent use…
· (Line 33) … ultrasound radiation, etc.
· (Line 48) Structures of N-aryl-4-arylpyridines…
· (Line 75) … by Lee, Y. R. et al.
· (Table 1) 0C should be either oC or °C.
· (Line 246) 1074/72 cm-1 is likely … (space is needed)
· (Figure 7) % Yield should be changed to Yield (%) to match the reaction time format on the right side of the figure.
· (Page 15/Line 282) An extra blank page.
· (Line 304) There is no section 0 in the manuscript.
· (Scheme 5) Please clean up the structure of 12.
· (Line 313) Synthesis of N-aryl substituted…
20. (Supporting Information) 1) Please provide some cleaner NMR spectra or at least label the solvent peaks and impurities, but 2) please clean up some of the spectra as the software was picking up too many peaks from the data.
Comments on the Quality of English Language
Please see attached review.
